# Dietary Supplementation with Monosodium Glutamate Suppresses Chemotherapy-Induced Downregulation of the T1R3 Taste Receptor Subunit in Head and Neck Cancer Patients

**DOI:** 10.3390/nu13092921

**Published:** 2021-08-24

**Authors:** Hitoshi Shono, Rie Tsutsumi, Kana Beppu, Rina Matsushima, Suzuno Watanabe, Chisa Fujimoto, Ryo Kanamura, Hiroki Ohnishi, Eiji Kondo, Takahiro Azuma, Go Sato, Misako Kawai, Hideki Matsumoto, Yoshiaki Kitamura, Hiroshi Sakaue, Noriaki Takeda

**Affiliations:** 1Department of Otolaryngology, Institute of Biomedical Sciences, Tokushima University, Tokushima 770-8503, Japan; hshono722@gmail.com (H.S.); chisa0427@hotmail.co.jp (C.F.); ryo.kanamura@tokushima-u.ac.jp (R.K.); wahre_lied@hotmail.co.jp (H.O.); siverianj@yahoo.co.jp (E.K.); azuma.takahiro@tokushima-u.ac.jp (T.A.); go-sato@tokushima-u.ac.jp (G.S.); ykitamura@tokushima-u.ac.jp (Y.K.); takeda@tokushima-u.ac.jp (N.T.); 2Department of Nutrition and Metabolism, Institute of Biomedical Sciences, Tokushima University, Tokushima 770-8503, Japan; c201941004@tokushima-u.ac.jp (K.B.); mtsmrn1207@gmail.com (R.M.); sumomomo071218@gmail.com (S.W.); hsakaue@tokushima-u.ac.jp (H.S.); 3Frontier Research Laboratories, Institute for Innovation, Ajinomoto Co. Inc., Kawasaki 210-8681, Japan; mmmmkawai@yahoo.co.jp (M.K.); hideki_matsumoto@ajinomoto.com (H.M.)

**Keywords:** monosodium glutamate, T1R3 genes, head and neck cancer, dysgeusia

## Abstract

(Background) We investigated the effect of dietary supplementation with monosodium glutamate (MSG) on chemotherapy-induced downregulation of the T1R3 taste receptor subunit expression in the tongue of patients with advanced head and neck cancer. (Methods) Patients undergoing two rounds of chemoradiotherapy were randomly allocated to a control or intervention group (dietary supplementation with MSG at 2.7 g/day during the second round of chemotherapy). The relative expression of T1R3, a subunit of both umami and sweet taste receptors, in the tongue was assessed by quantitative polymerase chain reaction analysis. Dysgeusia was assessed with a visual analog scale and daily energy intake was evaluated. (Results) T1R3 expression levels in the tongue, taste sensitivity, and daily energy intake were significantly reduced after the first round of chemotherapy compared with before treatment. Furthermore, these parameters significantly decreased after the second round of chemotherapy, but the extent of decrease was significantly attenuated in the MSG group compared with the control group. (Conclusions) MSG supplementation suppresses chemotherapy-induced dysgeusia, possibly due to the inhibition of the T1R3-containing taste receptor downregulation in the tongue, thereby increasing energy intake in patients with advanced head and neck cancer.

## 1. Introduction

Most cases of head and neck cancer include squamous cell carcinoma of the oral cavity, pharynx, or larynx. Patients usually receive chemoradiotherapy (CRT) as the primary treatment to achieve organ preservation. However, CRT often has adverse effects, such as stomatitis, dysgeusia, and nausea, in individuals with advanced head and neck cancer. Such adverse effects can result in decreased oral food intake, leading to malnutrition and interruption of CRT, potentially reducing its efficacy [1].

The basic taste modalities are sweet, sour, bitter, salty, and umami, which are perceived by specific taste receptors in the taste buds on the tongue [2]. The umami and sweet taste receptors are heterodimeric proteins composed of T1R1 and T1R3 and T1R2 and T1R3 subunits, respectively [3,4], whereas bitter taste receptors are monomers of the 25 types of T2R subunits [5]. The perception of sweet and bitter tastes declines in cancer patients undergoing chemotherapy, including those with head and neck cancer [6]. Chemotherapy is associated with a decrease in T1R3 abundance in the tongue of patients with head and neck cancer [7]. Given that the expression of taste receptor genes in the tongue is related to taste sensation [8], chemotherapy-induced taste impairment may in part result from the downregulation of T1R3 expression in the tongue.

The monosodium salt of L-glutamate (MSG) is an umami compound used worldwide as a seasoning and is detected by umami taste receptors in the tongue [9]. Stimulation of taste buds with MSG results in upregulation of T1R1 and T1R3 expression levels in the tongue of healthy individuals [10]. Here, we have hypothesized that stimulating T1R3 by MSG might support maintaining T1R3 gene expression, resulting in prevent dysgeusia with improvement of nutritional status. To improve the nutritional supportive care for patients undergoing cancer treatment, we examined the effect of dietary supplementation with MSG on the chemotherapy-induced downregulation of T1R3 expression in the tongue of patients with advanced head and neck cancer. MSG was added to rice at each meal three times a day for 7 days. A mucosal scraping sample was obtained from the foliate papillae on the tongue of the study patients for assessing T1R3 abundance using real-time polymerase chain reaction (PCR) analysis. In addition, we examined the effects of dietary supplementation with MSG on the chemotherapy-induced impairment of taste and reduction in daily energy intake in the same patients.

## 2. Materials and Methods

### 2.1. Subjects

The study was approved by the Tokushima University Hospital Committee for Medical Ethics (first registration date; 1 August 2011, approval number; 1235, UMIN: 00005885), and written informed consent was obtained from each patient prior to enrollment. All methods were performed in accordance with the relevant guidelines and regulations stated by the Institutional Ethics Committee Tokushima University Hospital, Japan. A total of 51 patients (41 men and 10 women) with advanced head and neck cancer, with age ranging between 37–82 years (mean ± standard deviation: 63.2 ± 9.3 years), and undergoing CRT as the first-line treatment were enrolled in the study. All patients received radiation therapy at a dose of 60–70 Gy over 12 weeks. They also received either cisplatin (80–100 mg/m^2^, two or three times) or cisplatin (80 mg/m^2^, twice) plus 5-fluorouracil (800 mg/m^2^), with a 3-week washout period. 

### 2.2. Dietary MSG Supplementation

After the first round of chemotherapy, the patients were randomly divided into two groups. The MSG group comprised 25 patients (20 men and 5 women, age: 37–82 years [64.2 ± 10.6 years]) with laryngeal cancer (*n* = 2), hypopharyngeal cancer (*n* = 5), pharyngeal cancer (*n* = 10), maxillary cancer (*n* = 1), other paranasal cancers (*n* = 4), or other head and neck cancers (*n* = 3). MSG powder (Ajinomoto, Tokyo, Japan) was added directly to rice as a seasoning during the second round of chemotherapy, with the powder being sprinkled over the rice by clinic staff to ensure compliance. The powder contained 0.9 g MSG for consumption three times per day for the first 7 days. The control group comprised 26 patients (21 men and 5 women, age: 49–80 years [62.4 ± 8.01 years]) and included individuals with laryngeal cancer (*n* = 1), hypopharyngeal cancer (*n* = 4), pharyngeal cancer (*n* = 15), maxillary cancer (*n* = 2), other paranasal cancer (*n* = 1), or other head and neck cancers (*n* = 3). Patients in the control group did not receive MSG supplementation.

### 2.3. RT-PCR Analysis

The surface of the foliate papillae on the tongue of patients was scraped with a small spatula after local anesthesia with 4% lidocaine to collect a sample of the lingual mucosa [11]. Scraping was performed immediately before and 1 week after the first and second doses of chemotherapy. All tissue scrapings were immediately mixed with RNAlater solution (Ambion, Austin, TX, USA) and the RNA was extracted using an RNAqueous phenol-free RNA isolation kit (Ambion) and amplified using the CellAmp Whole Transcriptome Amplification Kit Version 2 (Takara Bio, Shiga, Japan). Total RNA (1 µg) was reverse-transcribed in a final volume of 20 µL using a Primescript RT Reagent Kit (Takara Bio). The resulting cDNA (50 ng) was subjected to real-time PCR using specific primers in a final volume of 10 µL using a StepOnePlus Real-Time PCR System (Life Technologies, Waltham, MA, USA). The sequences of the primer sets used (forward and reverse, respectively) were 5′–TTCCCCCAGTACGTGAAGAC–3′ and 5′–CAGAGAACGTCTGGTGGTGA–3′ for the human T1R3, and 5′–GAAATCCCATCACCATCTTCCAGG-3′ and 5′–GAGCCCCAGCCTTCTCCATG–3′ for the human glyceraldehyde-3-phosphate dehydrogenase (*GAPDH*) (Invitrogen, Waltham, MA, USA). The PCR products were quantified by fit-point analysis and the expression of T1R3 was normalized with that of *GAPDH*.

### 2.4. VAS of Taste Sensitivity

Patients were asked to mark a score corresponding to their taste sensitivity using a visual scale [12] from 0 (not sensitive at all) to 100 (no problem of taste sensation) immediately before and 1 week after each of the first two doses of chemotherapy. Each of the five tastes and total sensitivity were assessed and the total score was adopted as taste sensitivity.

### 2.5. Daily Energy Intake

Patients were provided hospital food service. Dietary intake was calculated by a registered dietician by assessment of a 24-h dietary record obtained over 7 days. Average energy intake over 7 days was compared during the weeks before and after the first and second doses of chemotherapy.

### 2.6. Statistical Analysis

Data are presented as means ± standard deviation. A paired *t*-test was performed after assessing data normality. Multiple comparisons were performed using the Kruskal–Wallis test and two-way ANOVA analysis of variance. If an overall significant difference was detected, the Tukey–Kramer test was applied to identify pairs showing significant differences. Correlation analysis was performed using the Spearman correlation test. All statistical analysis and graph generation were conducted with the JMP software (SAS Institute, Tokyo, Japan) or PRISM 7 software (GraphPad Software, San Diego, CA, USA). Results with *p* < 0.05 were considered statistically significant.

## 3. Results

### 3.1. Characteristics of the Study Participants

A total of 51 patients with advanced head and neck cancer (41 men and 10 women, age: 37–82 years [mean age ± standard deviation: 63.2 ± 9.3 years]) undergoing CRT were enrolled in the study. In the first round of chemotherapy during CRT, MSG was not added to the rice fed to the patients. The MSG group comprised 25 patients (20 men and 5 women, age: 37–82 years [64.2 ± 10.6 years]) with laryngeal cancer (*n* = 2), hypopharyngeal cancer (*n* = 5), pharyngeal cancer (*n* = 10), maxillary cancer (*n* = 1), other paranasal cancers (*n* = 4), or other head and neck cancers (*n* = 3). The powder contained 0.9 g MSG for consumption three times per day for the first 7 days. The control group comprised 26 patients (21 men and 5 women, age: 49–80 years [62.4 ± 8.01 years]) and included individuals with laryngeal cancer (*n* = 1), hypopharyngeal cancer (*n* = 4), pharyngeal cancer (*n* = 15), maxillary cancer (*n* = 2), other paranasal cancer (*n* = 1), or other head and neck cancers (*n* = 3). Patients in the control group did not receive MSG supplementation.

### 3.2. Change of Lingual T1R3 Gene Expression 

One week after the first dose of chemotherapy, the expression of T1R3 gene expression in the tongue decreased significantly compared with that measured before chemotherapy (* *p* < 0.05 before vs. after chemotherapy in both the control and MSG groups) (Figure 1). Measurement of T1R3 levels immediately before the second round of chemotherapy revealed that its expression had recovered in both groups (*p* = 0.986 in the control group and *p* = 0.783 in the MSG group as compared before the first round of chemotherapy vs. before the second round). Patients in the MSG group were fed rice with MSG three times a day (2.7 g/day) for the first 7 days during the second round of chemotherapy, whereas patients in the control group received no supplementation. One week after the second dose of chemotherapy, the T1R3 expression in the tongue again decreased significantly in both groups of patients compared with before the second dose. However, the T1R3 levels in the MSG group were significantly higher than that in the control group after the second dose (‡ *p* < 0.05, control group vs. MSG group).

### 3.3. Improvement of Visual Analog Scale 

One week after the first dose of chemotherapy, the visual analog scale (VAS) score for taste sensitivity decreased significantly in both control and MSG groups of patients compared with immediately before chemotherapy (* *p* < 0.05 before vs. after chemotherapy in both the control and MSG groups) (Figure 2). The score returned to baseline immediately before the second round of chemotherapy but again decreased at 1 week after the second dose in each group. However, the VAS score was significantly higher in the MSG group than in the control group at 1 week after the second dose of chemotherapy (‡ *p* < 0.05, control group vs. MSG group). Daily energy intake in both the control and MSG groups of patients decreased significantly at 1 week after compared with before the first dose of chemotherapy (* *p* < 0.05, before vs. after chemotherapy in both the control and MSG groups) (Figure 3). Although daily energy intake recovered immediately before the second round of chemotherapy, it again decreased significantly in both groups at 1 week after the second dose. Nevertheless, the daily energy intake in the MSG group was significantly higher than that in the control group at 1 week after the second dose of chemotherapy (‡ *p* < 0.05, control group vs. MSG group). 

### 3.4. Correlation between T1R3 Gene and VAS

The abundance of T1R3 expression showed a significant positive correlation with the VAS score of taste sensitivity in the head and neck cancer patients of both control and MSG groups throughout CRT (r = 0.404, *p* < 0.01) (Figure 4a). The correlation coefficient was lower for patients in the two groups after the second dose of chemotherapy (r = 0.370, *p* < 0.05) (Figure 4b). The distribution of points at this time also differed between the control group, in which taste was impaired without any intervention, and the MSG group, in which taste was maintained by dietary supplementation with MSG. A significant positive correlation was also detected between the VAS score of taste sensitivity and daily energy intake after the second dose of chemotherapy in patients of the two groups (r = 0.289, *p* < 0.05) (Figure 5).

## 4. Discussion

Patients with advanced head and neck cancer often experience adverse effects, including dysgeusia during CRT. Given that such adverse effects can result in reduced food intake, leading to malnutrition and potential CRT interruption, effective supportive care is needed during cancer treatment. In the present study, we observed that while the first round of chemotherapy (cisplatin only or a combination of cisplatin and 5-fluorouracil) during CRT resulted in a decrease in T1R3 expression levels in the lingual mucosa and taste sensitivity in patients with advanced head and neck cancer, these effects were attenuated by dietary supplementation with MSG during the second round of chemotherapy, as shown in the graphic abstract in Figure 6. This is the first report that maximizes the characteristics of MSG and reveals that nutritional components are effective for dysgeusia that occurs during chemotherapy.

Herein, we focused on the effects of MSG and not on those of other umami compounds, such as the 5′-ribonucleotides guanosine monophosphate and inosine monophosphate, as our preliminary work showed that MSG was the only compound to increase T1R3 expression in the tongue of mice (unpublished data). In addition, we focused on T1R3 because chemotherapy (but not radiation therapy) reportedly reduces T1R3 expression in the tongue of head and neck cancer patients without altering T1R1 or T1R2 expression, while it increases the expression of the T2R proteins, which sense bitter taste [7]. Given that glucose control is important in cancer patients, we did not examine the effects of sweet compounds, even though T1R3 is a common subunit of sweet and umami taste receptors [4].

Previous studies have also shown that MSG induces upregulation of T1R3 expression [10,13]. A study in healthy volunteers reported that MSG induced the expression of both T1R1 and T1R3 in tongue [10]. Oral administration of MSG also increased the gastrointestinal mRNA and protein levels of T1R1 and T1R3 in piglets [13]. We have recently published that T1R3 mRNA were increased in healthy subject whose dairy diet suggested unbalanced and low glutamate [14]. Thus, we hypothesized malnutrition or poor dietary intake decreased T1R3 and MSG could improve these conditions with T1R3 gene expression increases. Although the precise mechanisms of these effects remain unclear, it is likely that the interaction of MSG with T1R1/T1R3 umami receptors in the tongue of patients with head and neck cancer activates intracellular signals that elicit the expression of T1R3, thereby compensating for the inhibitory effect of chemotherapy. We detected only a 1.2-fold increase in T1R3 gene expression by MSG, which might not significantly affect taste perception. However, we have previously reported that a decrease in taste under chemotherapy correlates with a decrease in the expression of taste receptor genes, and at this time, a decrease in gene expression of only about 20% occurred [7].

Dietary zinc is also believed to play an important role in dysgeusia and taste receptor expression. The turnover of taste buds is prolonged and the expression of T2R is reduced in zinc-deficient rats [15]. Although not addressed in the present study, zinc deficiency due to undernutrition may contribute to T1R3 downregulation during chemotherapy.

The first and second rounds of chemotherapy resulted in a decreased VAS score of taste sensitivity and daily energy intake in the study patients; effects that were attenuated by MSG supplementation after the second chemotherapy dose. Moreover, after the second round of chemotherapy, a significant correlation was notable between the VAS score of taste sensitivity and both T1R3 abundance in the tongue and daily energy intake for patients in the MSG and control groups. As shown in Figure 2, in the intervention group in which the T1R3 gene expression was significantly increased by MSG, the perceived taste perception was improved, resulting in an increase in the dietary intake, as shown Figure 3. This is also supported in Figure 4. We believe that the increase in the T1R3 gene by MSG is medically very important because decreased dietary intake leads to weight loss, and excessive weight loss makes it difficult to continue treatment, and thus, leads to a poor prognosis [1]. In mice, umami receptors are expressed on smooth muscle cells in the stomach [16]. Dietary MSG reportedly increases the secretion of saliva from parotid glands [17], the secretion of gastric juice [18], and gastrointestinal peristalsis [19] by stimulating gastric taste receptors, thus promoting food ingestion and digestion [20]. In elderly and young individuals, salivary IgA secretion is stimulated by T1R3 activation [21], and dietary supplementation with MSG improved nutritional status and secretion of gastric juice in hospitalized elderly patients [22].

In the present study, to assess the effects of dietary supplementation with MSG on dysgeusia and food intake, only patients with head and neck cancer who consumed hospital meals throughout the hospitalization period were enrolled. Under these controlled conditions, we found that chemotherapy-induced taste impairment was accompanied by a reduction in daily energy intake. Our results suggest that dietary supplementation with MSG suppressed chemotherapy-induced dysgeusia by attenuating the associated downregulation of umami and sweet taste receptors in the tongue. Malnutrition is associated with reduced overall survival in patients with head and neck cancer [23], and energy intake is a strong independent predictor of survival [24,25]. In addition, glutamate is known to prevent intestinal atrophy [26]. Chemotherapy increases glutamate transport to protect from intestinal mucosa [27]. These publications lead to suggest glutamate also could protect oral mucosity. Decreased saliva secretion is also a serious problem during CRT, but MSG has contributed to the promotion of saliva secretion during meals [28], which can be expected to improve QOL, such as eating delicious meals. Therefore, dietary supplementation with MSG may not only improve taste-related quality of life but also ameliorate malnutrition and, thereby, reduce mortality with improved QOL in patients with advanced head and neck cancer treated with CRT. 

There are several limitations to the present study. First, to quantify T1R3 mRNA expression levels, we used scraped samples. Since it is not practical to perform a biopsy of the human tongue, this time we extracted mRNA from a sample that rubbed the surface of the tongue. Since T1R3 is a GPCR and is expressed on the tongue surface and we previously have also measured histamine receptors from the nasal mucosa [11], it is considered possible to measure. In addition, taste sensation was assessed only with a VAS. However, a more accurate taste test, such as a whole-mouth gustatory test, would have imposed too great a burden on the study patients, although T1R3 expression in the tongue is significantly correlated with threshold levels in the whole-mouth gustatory test [7]. Second, the number of patients was relatively small, and the study was designed as a cohort study. Thus, further randomized clinical trials are needed to confirm our findings. We did confirm that the control and intervention groups were not biased regarding chemotherapy with cisplatin alone vs. cisplatin plus 5-fluorouracil after random assignment. Third, the dose-response relations for the effects of MSG were not investigated. MSG supplementation was fixed at 2.7 g/day because supplementation with 0.6% MSG per meal (1.2 g of MSG) was previously shown to increase appetite in elderly hospitalized patients and improve meal satisfaction [22]. Supplementation with 0.6% MSG (daily consumption of 3 g of MSG) was also found to enhance palatability in the context of the French diet and could be continued for long periods in both young and elderly individuals [29]. The U.S. Food and Drug Administration recommends that no more than 10 g of MSG should be consumed at one time.

In conclusion, this study showed that dietary supplementation with MSG suppresses chemotherapy-induced dysgeusia by attenuating the downregulation of T1R3 expression in the tongue of patients with advanced head and neck cancer treated with CRT. As a result, MSG intake also suppresses the chemotherapy-induced decrease in daily energy intake in these patients. Taken together, our results provide a compelling evidence that dietary supplementation with MSG during CRT may be a promising nutritional intervention to improve prognosis in patients with advanced head and neck cancer undergoing CRT.

## 5. Conclusions

Dietary supplementation with MSG during CRT may be a promising nutritional intervention to improve prognosis in patients with advanced head and neck cancer undergoing CRT.

## 6. Patents

Patent number 2017-528976 resulted from this work.

## Figures and Tables

**Figure 1 nutrients-13-02921-f001:**
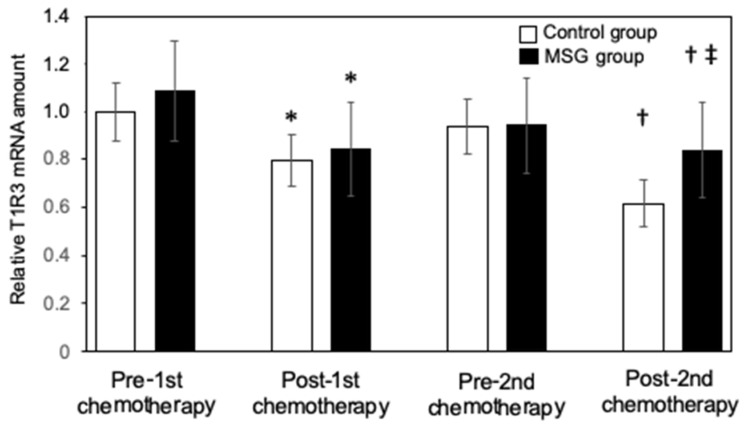
Effect of dietary monosodium glutamate (MSG) supplementation on chemotherapy-induced downregulation of T1R3 expression in the tongues of patients with head and neck cancer. Lingual T1R3 expression levels were measured in patients randomly assigned to the control (*n* = 26) and MSG (*n* = 25) groups immediately before and 1 week after the first and second doses of chemotherapy. Data are represented as the mean ± standard deviation and are expressed relative to the pre-first chemotherapy value for the control group. * *p* < 0.05 vs. the corresponding value for the pre-first chemotherapy, † *p* < 0.05 vs. the corresponding value for pre-second chemotherapy, ‡ *p* < 0.05 vs. the corresponding value for the control group as determined by the Kruskal–Wallis test and two-way ANOVA analysis of variance. If an overall significant difference was detected, the Tukey–Kramer test was applied to identify pairs showing significant differences.

**Figure 2 nutrients-13-02921-f002:**
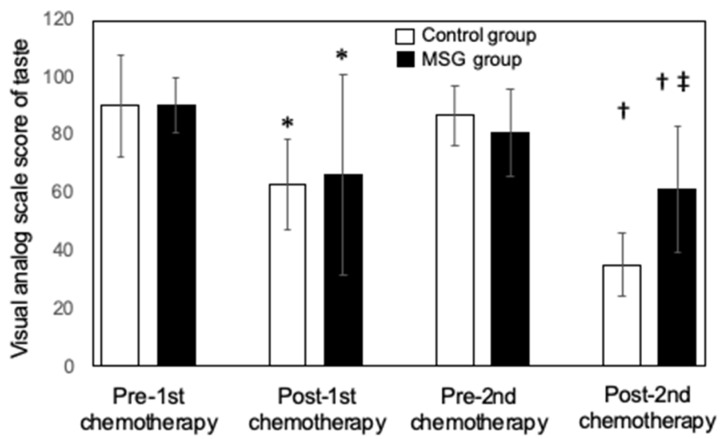
Effect of dietary monosodium glutamate (MSG) supplementation on chemotherapy-induced taste impairment in patients with head and neck cancer. The visual analog scale (VAS) score of taste sensitivity was measured in patients randomly assigned to the control (*n* = 26) and MSG (*n* = 25) groups immediately before and 1 week after the first and second doses of chemotherapy. Data are represented as the mean ± standard deviation. * *p* < 0.05 vs. the corresponding value for pre-first chemotherapy, † *p* < 0.05 vs. the corresponding value for pre-second chemotherapy, ‡ *p* < 0.05 vs. the corresponding value for the control group. Multiple comparison was performed by the Kruskal–Wallis test and two-way ANOVA analysis of variance. If an overall significant difference was detected, the Tukey–Kramer test was applied to identify pairs showing significant differences.

**Figure 3 nutrients-13-02921-f003:**
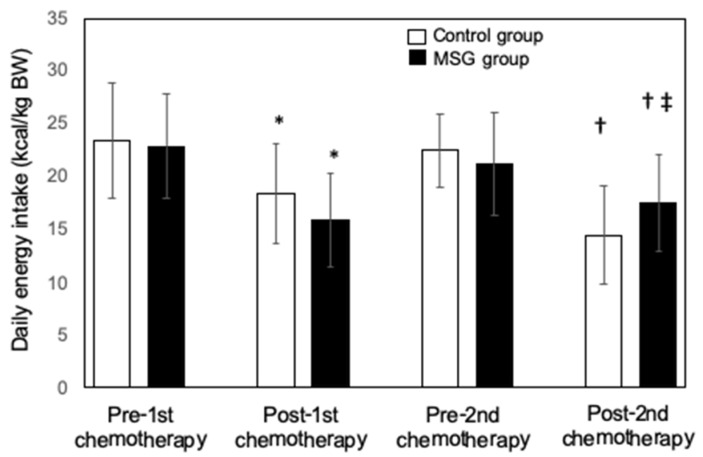
Effect of dietary monosodium glutamate (MSG) supplementation on chemotherapy-induced reduction in daily energy intake in patients with head and neck cancer. Daily energy intake (kilocalories per kilogram of body weight) was measured in patients randomly assigned to the control (*n* = 26) and MSG (*n* = 25) groups during the week before and the week after the first and second doses of chemotherapy. Data are represented as the mean ± standard deviation. * *p* < 0.05 vs. the corresponding value for pre-first chemotherapy, † *p* < 0.05 vs. the corresponding value for pre-second chemotherapy, ‡ *p* < 0.05 vs. the corresponding value for the control group. Multiple comparison was performed by the Kruskal–Wallis test and two-way ANOVA analysis of variance. If an overall significant difference was detected, the Tukey–Kramer test was applied to identify pairs showing significant differences.

**Figure 4 nutrients-13-02921-f004:**
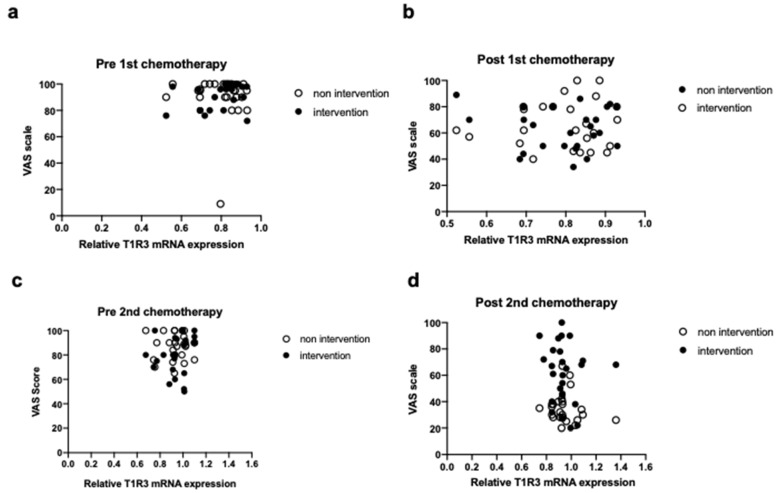
Correlation between the visual analog scale (VAS) score of taste sensitivity and lingual T1R3 abundance in patients with head and neck cancer who did or did not receive monosodium glutamate (MSG) dietary supplementation during chemoradiotherapy. (**a**). Spearman correlation analysis for patients in both control and MSG groups before first chemotherapy. (**b**). Spearman correlation analysis for patients in both control and MSG groups after first chemotherapy. (**c**). Spearman correlation analysis for patients in both control and MSG groups before second chemotherapy. (**d**). Spearman correlation analysis for patients in both control and MSG groups after second chemotherapy. The mRNA data in all panels are expressed relative to the mean value for the control group before the first dose of chemotherapy. Open circle: non-intervention control group, closed circle: intervention MSG group.

**Figure 5 nutrients-13-02921-f005:**
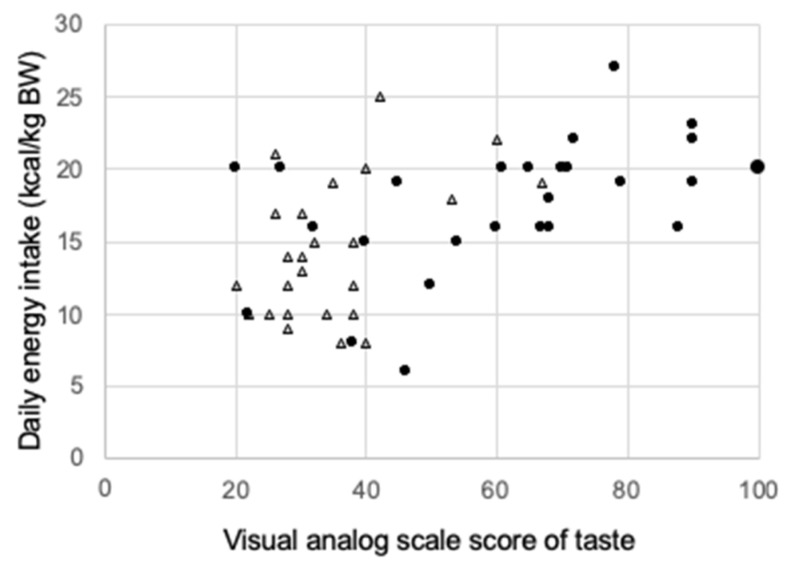
Correlation between daily energy intake and the visual analog scale (VAS) score of taste sensitivity after the second dose of chemotherapy in patients with head and neck cancer upon monosodium glutamate (MSG) dietary supplementation. Opened triangle; non intervention group, Closed circled; intervention group.

**Figure 6 nutrients-13-02921-f006:**
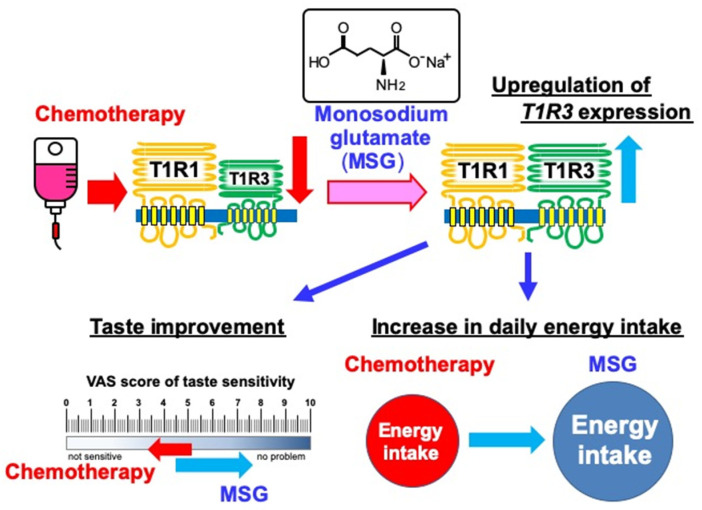
Chemotherapy with MSG supplementation. MSG; monosodium glutamate.

## Data Availability

The data presented in this study are available on request from the corresponding author.

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
