# Peer review of "Dietary Supplementation with Monosodium Glutamate Suppresses Chemotherapy-Induced Downregulation of the T1R3 Taste Receptor Subunit in Head and Neck Cancer Patients"

_nutrients, 2021, doi:10.3390/nu13092921_

Round 1
Reviewer 1 Report
The article with the title “Dietary supplementation with monosodium glutamate suppresses chemotherapy-in- duced downregulation of the T1R3 taste receptor subunit in head and neck cancer patients” is in generally interesting, but I would offer these comments to the investigators:
1) Some minor grammatical errors.
2) The reference list is quite old. Authors should focus on recent papers and papers older than three years should except for an overriding purpose
3) The authors used only one concentration of MSG. I believe that subgroups with different doses of MSG would evaluate a dose dependent manner. Thus, a more solid correlation would establish between MSG and T1R3 expression.
4) The age of the patients affected the expression of T1R3. Please explain.
5) Please explain if the expression of other taste receptors mRNA levels were measured?
6) Did the authors compare the effect of MSG on the T1R3 expression with other known agents with the same effect on T1R3?

Author Response
Thank you so much for the review and helpful suggestions.
Comments;
1) Some minor grammatical errors.
Response;
Since our manuscript were edited by Editage editing company, we asked edit again and have attach certificate.
Comments;
2) The reference list is quite old. Authors should focus on recent papers and papers older than three years should except for an overriding purpose
Response;
We have replaced some references with new ones and added recent publications. However, there are few papers that use MSG and umami for treatment, so that old ones are inevitably left.
Comments;
3) The authors used only one concentration of MSG. I believe that subgroups with different doses of MSG would evaluate a dose dependent manner. Thus, a more solid correlation would establish between MSG and T1R3 expression.
Response
In fact, we also made a 1/10 amount of sprinkle of 0.09g per sprinkle, but we couldn't get enough interventions. However, in healthy subjects, a crossover study was conducted at 2.7 g / day and 0.27 g / day, and an increase in the expression of the T1R3 gene was observed only with the intervention of 2.7 g / day. The results of the MSG intervention trial in healthy individuals have been published, so I wrote them as a discussion in the literature and added them to the citations (Line263-267).
Comments
4) The age of the patients affected the expression of T1R3. Please explain.
Response;
we have add limitation about age on discussion as below;
Therefore, the difference in the effect of MSG intervention depending on age and gender has not been fully examined. However, there was no large standard deviation in patient age, and there was no age difference between the two groups, interventional and non-intervention. (Line326-329)
commnets
5) Please explain if the expression of other taste receptors mRNA levels were measured?
response;
although we have assessed T1R1 and T1R2 gene expressions, we did not see any significant change. so we have added about this on result and discussion (Line155-157,259-263 ).
6) Did the authors compare the effect of MSG on the T1R3 expression with other known agents with the same effect on T1R3?
In animal tests, inosinic acid and guanylic acid, which are known as umami components like MSG, did not increase the expression of the T1R3 gene. Also, as far as we know, no drug that increases T1R3 has been reported.

Reviewer 2 Report
This manuscript deals with an important subject in cancer therapy: side effects of radio- and chemotherapy. The authors analyze loss of taste after chemoradiotherapy of head and neck cancer patients and analyze expression of taste receptor in tongue as well as energy expedition in these patients. They concluded that dietary supplementation with monosodium glutamate (MSG) inhibited dysgeusia resulted from chemotherapy with cisplatin and 5-fluorouracil associated with radiation due to preventing T1R3-containing taste receptor downregulation in the tongue of the patients. The authors associated their observation with an increased energy intake in MSG-receiving patients. These results are interesting and clearly presented, but altogether they do not constitute a set of data justifying their publication in Nutrients. They are a combination of real-time PCR with visual analog scale. I suggest the Authors answer the following questions:
- How does act MSG on cancer and chemotherapy-free patients in the context of parameters they investigated?
- Are results of mRNA level sufficient to conclude about expression of the T1R receptors?
- Does statistical significance the Authors observed justify its biological/medical relevance?
In summary, this manuscript in its present form is below standard expected from manuscripts to be published in Nutrients – it may be, after reduction, considered as a short communication or preliminary results.
Author Response
Comments
- How does act MSG on cancer and chemotherapy-free patients in the context of parameters they investigated?
Response
Although we have not determined detailed mechanism of MSG on lingual taste receptor increase, we have recently published that T1R3 mRNA were increased in healthy subject whose dairy diet include low glutamate, high sugar and high salt. Thus, we hypothesized malnutrition or poor dietary intake decreased T1R3 and MSG could improve these conditions with T1R3 gene expression increases. We described about this on discussion as below.
Line 263-267;
We have recently published that T1R3 mRNA were increased in healthy subject whose dairy diet include low glutamate, high sugar and high salt [26]. Thus, we hypothesized malnutrition or poor dietary intake decreased T1R3 and MSG could improve these conditions with T1R3 gene expression increases.
Commnets
- Are results of mRNA level sufficient to conclude about expression of the T1R receptors?
Response
Thank you for your pointing out. Perhaps you point out that a 1.2-fold increase in gene expression does not significantly affect taste perception. However, we have previously reported that a decrease in taste under chemotherapy correlates with a decrease in the expression of taste receptor genes, and at this time, a decrease in gene expression of about 20% occurred. For this reason, we also added to the consideration of the rate of increase in gene expression.
Line 271-276;
Although we detected only 1.2-fold increase in T1R3 gene expression by MSG, and might not significantly affect taste perception. However, we have previously reported that a decrease in taste under chemotherapy correlates with a decrease in the expression of taste receptor genes, and at this time, a decrease in gene expression of only about 20% occurred [7].
Comments
- Does statistical significance the Authors observed justify its biological/medical relevance?
Response
Response:
As shown in Figure2, in the intervention group in which the T1R3 gene expression was significantly increased by MSG, the perceived taste perception was improved, resulting in increase in the dietary intake as shown Figure3. This is also supported in Figure4. We believe that the increase in the T1R3 gene by MSG is medically very important because decreased dietary intake leads to weight loss, and excessive weight loss makes it difficult to continue treatment and thus leads to a poor prognosis. Therefore, this point was further emphasized in the discussion.
LINE287-293
As shown in Figure 2, the perceived taste perception was improved in the intervention group, in which the expression of the T1R3 gene was significantly increased by MSG, resulting in turn in an increase in the dietary intake, as shown in Figure 3. This finding is also supported by the data illustrated in Figure 4. We believe that the MSG-induced increase in the expression of the T1R3 gene is very important clinically, as the decreased dietary intake leads to weight loss that hinders the continuation of treatment and thus leads to poor prognosis [1].

Reviewer 3 Report
The publication "Dietary supplementation with monosodium glutamate suppresses chemotherapy-induced downregulation of the T1R3 taste receptor subunit in head and neck cancer patients" is an interesting scientific study. Abstract is properly spelled. It would be advisable to include a graphic abstract, so popular in recent years. The introduction, materials and methods, and the results are compiled in a concise manner. The results are presented in the form of bar graphs. The short discussion also includes the comparison of the Authors' own results with data from references. The results of the publication are not only scientific, but also practical. They provide valuable tips for nutritionists working in cancer hospitals.
Author Response
Thank you for your kind and motivational comments for future research. As following your suggestion, we have added graphic abstract as Figure 6.

Reviewer 4 Report
The authors describe their work on the effects of dietary MSG supplementation on TIR3 taste receptors in patients with head and neck cancer undergoing chemotherapy. It was found that MSG suppresses chemotherapy induced dysgeusia due to the inhibition of the TIR3 containing taste receptor downregulation in the tongue, thus increasing energy intake in patients with advanced head and neck cancer. This is an interesting study. Appropriate methodology has been employed and the conclusions appear to be justified based on the data at hand. The authors also describe the limitations of their study very well. I have a few recommendations for consideration.
- Introduction. Please provide a clear hypothesis to be tested in the study..
- Results. Figs. 1-3 are very confusing. For example incorrect use of symbols to indicate statistical differences. I presume open bars are control and filled bars are + MSG. Also, the comparisons being undertaken for these data as well as the interpretation of the data need to be clearly defined and described in the results section.
- Discussion. The authors need to elaborate and emphasize the novelty aspect of their work.
Author Response
Comments:
- Introduction. Please provide a clear hypothesis to be tested in the study.
Response:
Line50-52:
Here, we have hypothysized that stimulating T1R3 by MSG might support to maintain T1R3 gene expression resulting inprevent dysgeusia with improvement of nutritional status.
- Results. Figs. 1-3 are very confusing. For example incorrect use of symbols to indicate statistical differences. I presume open bars are control and filled bars are + MSG. Also, the comparisons being undertaken for these data as well as the interpretation of the data need to be clearly defined and described in the results section.
Respons
Thank you so much for your suggestion. We changed open bar are control and closed bar to MSG. In addition, we have added more detailed information in result section as hilighted.
One week after the first dose of chemotherapy, the visual analog scale (VAS) score for taste sensitivity decreased significantly in both control and MSG groups of patients compared with immediately before chemotherapy (*p < 0.05 before vs after chemotherapy in both the control and MSG groups ) (Figure 2). The score returned to baseline immediately before the second round of chemotherapy but again decreased at 1 week after the second dose in each group. However, the VAS score was significantly higher in the MSG group than in the control group at 1 week after the second dose of chemotherapy (#p < 0.05, , control group vs MSG group). Daily energy intake in both the control and MSG groups of patients decreased significantly at 1 week after compared with before the first dose of chemotherapy (*p < 0.05, before vs after chemotherapy in both the control and MSG groups) (Figure 3). Although daily energy intake recovered immediately before the second round of chemotherapy, it again decreased significantly in both groups at 1 week after the second dose. Nevertheless, the daily energy intake in the MSG group was significantly higher than that in the control group at 1 week after the second dose of chemotherapy (#p < 0.05, control group vs MSG group).
- Discussion. The authors need to elaborate and emphasize the novelty aspect of their work.
Response;
As following your suggestion, we added several sentence to emphasize the novelty aspect of our study as below.
Line 242-244
This is the first report that maximizes the characteristics of MSG and reveals that nutritional components are effective for dysgeusia that occurs during chemotherapy.

Round 2
Reviewer 1 Report
The authors have answered all of my questions and the paper has been greatly improved
Author Response
Comments;
The authors have answered all of my questions and the paper has been greatly improved
Responce;
Thank you so much for your helpful review.
Reviewer 2 Report
My concerns have not been addressed specifically as I asked about effect of MSG on cancer-free, healthy subjects, which could be necessary to conclude about mechanism of taste change by MSG. This should be a sine qua non condition for considering this work. Second, gene expression is not limited to mRNA. The authors should perform analysis on the protein level (western blot, ELISA). Third, statistical significance is not the same as biological relevance.
Author Response
Thank you so much for your reviewing our manuscript again. We appreciate the valuable and helpful comments. Please find attached revised manuscript.
Comments;
My concerns have not been addressed specifically as I asked about effect of MSG on cancer-free, healthy subjects, which could be necessary to conclude about mechanism of taste change by MSG. This should be a sine qua non condition for considering this work.
Response;
We mentioned about our trial in cancer-free, healthy subjects because our paper was recently accepted on the effects of MSG on healthy subjects without cancer. (Line274-276)
Comments;
Second, gene expression is not limited to mRNA. The authors should perform analysis on the protein level (western blot, ELISA).
Response;
This study was a clinical trial in patients with head and neck cancer, and it was not feasible to perform an invasive biopsy to evaluate protein expression in those patients. Therefore, this time I added to the limitations (Line337-347)
Comments;
Third, statistical significance is not the same as biological relevance.
Response
As you pointed out, I agree that statistical significance is not the same as biological relevance. However, not only was a statistically significant relationship found in this study, but the dietary intake of patients is increased, and we believe that it has biologically important implications.

Reviewer 4 Report
The authors have addressed all my initial concerns and have adequately revised their manuscript. I have no further comments.
Author Response
Comments:
The authors have addressed all my initial concerns and have adequately revised their manuscript. I have no further comments.
Responce:
Thank you so much for your helpful comments.